# Riboflavin, a Potent Neuroprotective Vitamin: Focus on Flavivirus and Alphavirus Proteases

**DOI:** 10.3390/microorganisms10071331

**Published:** 2022-06-30

**Authors:** Raphael J. Eberle, Danilo S. Olivier, Marcos S. Amaral, Carolina C. Pacca, Mauricio L. Nogueira, Raghuvir K. Arni, Dieter Willbold, Monika A. Coronado

**Affiliations:** 1Institute of Biological Information Processing (IBI-7: Structural Biochemistry), Forschungszentrum Jülich GmbH, 52428 Jülich, Germany; d.willbold@fz-juelich.de; 2Institut für Physikalische Biologie, Heinrich-Heine-Universität Düsseldorf, Universitätsstraße, 40225 Düsseldorf, Germany; 3Center of Integrated Sciences, Campus Cimba, Federal University of Tocantins, Araguaína 77824-838, TO, Brazil; dolivier@gmail.com; 4Institute of Physics, Federal University of Mato Grosso do Sul, Campo Grande 79070-900, MS, Brazil; marcos.amaral@ufms.br; 5Instituto Superior de Educação Ceres, FACERES Medical School, São José do Rio Preto 15090-305, SP, Brazil; carolpacca@gmail.com; 6Laboratório de Pesquisas em Virologia, Departamento de Doenças Dermatológicas, Infecciosas e Parasitárias, Faculdade de Medicina de São José do Rio Preto-FAMERP, São José do Rio Preto 15090-000, SP, Brazil; mauricio.nogueira@edu.famerp.br; 7Department of Pathology, University of Texas Medical Branch, Galveston, TX 77550, USA; 8Multiuser Center for Biomolecular Innovation, Department of Physics, IBILCE, São Paulo State University, São Jose do Rio Preto 15054-000, SP, Brazil; raghuvir.arni@unesp.br; 9JuStruct: Jülich Centre for Structural Biology, Forchungszentrum Jülich, 52428 Jülich, Germany

**Keywords:** alphavirus, flavivirus, NS2B/NS3^pro^, nsP2^pro^, riboflavin, competitive inhibitor

## Abstract

Several neurotropic viruses are members of the flavivirus and alphavirus families. Infections caused by these viruses may cause long-term neurological sequelae in humans. The continuous emergence of infections caused by viruses around the world, such as the chikungunya virus (CHIKV) (Alphavirus genus), the zika virus (ZIKV) and the yellow fever virus (YFV) (both of the Flavivirus genus), warrants the development of new strategies to combat them. Our study demonstrates the inhibitory potential of the water-soluble vitamin riboflavin against NS2B/NS3^pro^ of ZIKV and YFV and nsP2^pro^ of CHIKV. Riboflavin presents a competitive inhibition mode with IC_50_ values in the medium µM range of 79.4 ± 5.0 µM for ZIKV NS2B/NS3^pro^ and 45.7 ± 2.9 μM for YFV NS2B/NS3^pro^. Against CHIKV nsP2^pro^, the vitamin showed a very strong effect (93 ± 5.7 nM). The determined dissociation constants (K_D_) are significantly below the threshold value of 30 µM. The ligand binding increases the thermal stability between 4 °C and 8 °C. Unexpectedly, riboflavin showed inhibiting activity against another viral protein; the molecule was also able to inhibit the viral entry of CHIKV. Molecular dynamics simulations indicated great stability of riboflavin in the protease active site, which validates the repurposing of riboflavin as a promising molecule in drug development against the viruses presented here.

## 1. Introduction

Neurotropic viruses that trigger neuronal dysfunctions are neuroinvasive or neurovirulent and infect hematopoietic cells, which serve as a platform to gain access to the central nervous system (CNS) via the blood supply [1,2]. To promote infection and migration (neurotropism), some neurotropic viruses use peripheral nerves, a second major route of CNS entry. Alpha- and flavivirus infections can cause various syndromes, ranging from benign febrile illnesses to severe systemic diseases with hemorrhagic manifestations or major organ involvement [3]. YFV, for instance, has unique hepatotropic properties that cause a clinically and pathologically distinct form of hepatitis with hemorrhagic diathesis [4]. In contrast, the live-attenuated vaccine (LAV) strain 17D does not cause viscerotropic disease and reversion to virulence is associated with neurotropic disease [5,6]. The viruses of both families are generally neurotropic, with the potential to cause fatal encephalitis associated with acute inflammation and widespread neuronal destruction as shown in Table 1.

Essential for normal cellular functions, growth and development, riboflavin (a water-soluble member of the B-vitamin family—B2) is an underestimated antioxidant that is the precursor of the essential coenzymes flavin adenine mononucleotide (FMN) and flavin adenine dinucleotide (FAD). Riboflavin consists of an isoalloxazine ring bound to a ribose side chain [17] (Appendix A) and is presented in bacteria and plants but absent in vertebrates [18]. Among others, it is responsible for the reduction in cellular oxidative stress [19], modulation of innate and immune responses [20], plays a role in protein folding in the endoplasmic reticulum [21] and exhibits anti-inflammatory properties [22]. Riboflavin excess does not result in toxicity since it is secreted in urine [17,23]. Several conditions can lead to riboflavin deficiency (ariboflavinosis), including lactation, diabetes mellitus, and phototherapy in infants, celiac sprue and more [17]. Ariboflavinosis leads to symptoms of neurodegeneration and peripheral neuropathy, as documented in several studies [23,24,25,26].

Our study is focused on proteases of the three neurotropic viruses ZIKV and YFV (Flaviviridae) and CHIKV (Alphaviridae) due to the large ZIKV, YFV and CHIKV outbreak in South America [27,28,29]. To date, there are no clinically safe drugs to combat these viral infections and there is an urgent need to identify promising lead compounds or drug candidates, as well as protective agents to prevent neurological damage. However, several studies have introduced compounds as potential ZIKV, YFV and CHIKV protease inhibitors and the examples are shown in Table 2.

ZIKV, YFV and CHIKV are enveloped, single stranded, positive-sense RNA viruses, transmitted primarily by arthropods (arboviruses) [27,28,29]. They enter host cells via viral glycoprotein receptor-mediated endocytosis and use the machinery of the infected cell to synthesize viral proteins and replicate their genome. The viral RNA genome encodes polyproteins containing structural proteins incorporated into the virions, and non-structural proteins that coordinate virus replication, assembly and modulation of host defense mechanisms [27,28,29]. The virus polyproteins are formed following proteolytic cleavage by viral protease activity [40,41], in the case of flaviviruses, the NS2B/NS3 serine protease complex (NS2B/NS3^pro^) and, in the case of alphaviruses, the nsP2 cysteine protease (nsP2^pro^). Both proteases represent attractive drug targets due to their essential role in the virus life cycle [40,41].

Gao et al. in 2014 demonstrated that riboflavin can interact with trypsin [42] and Baum et al. in 1996 showed that flavins inhibit human cytomegalovirus UL80 protease [43]. Based on these observations, we have utilized a combination of techniques to evaluate the inhibitory properties of riboflavin against ZIKV and YFV NS2B/NS3^pro^ and CHIKV nsP2^pro^. Our results indicated that riboflavin bound competitively to the active site of the tested proteases, with half maximal inhibition concentration (IC_50_) values in the moderate µM (ZIKV, YFV) and nM (CHIKV) range. Based on the fluorescence studies, we conclude that the interaction increases the thermal stability of these proteases. The visualizations of the binding interactions were obtained by in silico studies.

Our results indicate that riboflavin is a potentially interesting lead molecule for the development of inhibitors against the virus proteases studied. However, further studies are needed for the use of riboflavin as a protector against neurotropic viruses.

## 2. Materials and Methods

### 2.1. Cloning, Expression and Purification of Virus Proteases

The codon optimized cDNA encoding ZIKV NS2B/NS3^pro^ (GenBank protein accession number KU729217.2, Brazilian isolate BeH823339), YFV NS2B/NS3^pro^ (GenBank protein accession number AAY34247.1, isolate Angola/14FA/1971) and CHIKV nsP2^pro^ (GenBank protein accession number AAN05101.1, strain S27-African prototype) were cloned, expressed and purified as described previously [36,37,44]. Protease sample purity was verified by sodium dodecyl sulfate polyacrylamide gel electrophoresis (SDS-PAGE) under reducing conditions.

### 2.2. Inhibition Assay

The ZIKV NS2B/NS3^pro^, YFV NS2B/NS3^pro^, and CHIKV nsP2^pro^ activity assay was performed as described previously [36,37,44], using Pyr-Arg-Thr-Lys-Arg-AMC (BACHEM, Bubendorf, Switzerland) as the substrate for ZIKV NS2B/NS3^pro^, Boc-Gly-Arg-Arg-AMC (BACHEM, Bubendorf, Switzerland) for YFV NS2B/NS3^pro^ and DABCYL-Arg-Ala-Gly-Gly-Tyr-Ile-Phe-Ser-EDANS (BACHEM, Bubendorf, Switzerland) for CHIKV nsP2^pro^. The inhibition assay was performed in Corning 96-well plates (Merck, Darmstadt, Germany) for ZIKV NS2B/NS3^pro^ (3 nM), YFV NS2B/NS3^pro^ (6 nM), and CHIKV nsP2^pro^ (1 µM), separately. The proteins were incubated, separately, with riboflavin at the concentration of 0–240 µM (ZIKV), 0–200 μM (YFV) and 0–10 µM (CHIKV) and incubated for 1 h at RT. The measurement started by the addition of the substrate with a final concentration of 20 µM (ZIKV), 50 µM (YFV) or 5 µM (CHIKV) and, the fluorescence intensities were measured at 60 s intervals over 20 min at 37 °C using an Infinite 200 PRO plate reader (Tecan, Männedorf, Switzerland). The excitation and emission wavelengths were 380 nm and 460 nm (ZIKV), 380 nm and 465 nm (YFV) and 340 nm and 490 nm (CHIKV).

### 2.3. Determination of IC_50_, Inhibition Mode and LE

The half maximal inhibitor concentration (IC_50_) values of riboflavin were calculated using GraphPad Prism5 software (San Diego, CA, USA).

For the determination of the mode of inhibition, the assay was performed at different final concentrations of the inhibitors and substrates. First, 3 nM ZIKV NS2B/NS3^pro^ was pre-incubated with riboflavin at different concentrations for 60 min at RT (0–40 µM). CHIKV nsP2^pro^ at 1µM was incubated with 0-100 nM riboflavin and YFV NS2B/NS3^pro^ at 6 nM with 0–10 µM riboflavin. Subsequently, the reaction was initiated by addition of the corresponding concentration series of the substrate for ZIKV NS2B/NS3^pro^ (0–50 µM in 5 µM steps), YFV NS2B/NS3^pro^ (0–15 µM in 3 µM steps) and CHIKV nsP2^pro^ (0, 0.2, 0.4, 0.8, 1.0 and 2.0 µM). All measurements were performed in triplicate and data are presented as mean ± SD. The data analysis was performed using a Lineweaver–Burk plot; therefore, the reciprocal of velocity (1/V) vs the reciprocal of the substrate concentration (1/[S]) was compared [45,46].

To determine the ligand efficiency for riboflavin, Equation (1) was used [47].
(1.4∗pIC_50_)/N(1)
pIC_50_ was obtained from an online tool [48] and N is the number of all atoms, except hydrogen.

### 2.4. Fluorescence Spectroscopy

The intrinsic tryptophan (Trp) fluorescence of ZIKV NS2B/NS3^pro^, YFV NS2B/NS3^pro^ and CHIKV nsP2^pro^ was measured to determine the dissociation constant (K_D_) of riboflavin. A combined approach of a saturation–binding curve using a nonlinear least square fit procedure and a modified Hill equation was used to determine the K_D_ value; the experimental set up was as described previously [44]. The intrinsic Trp fluorescence was measured with a QuantaMaster40 spectrofluorometer (PTI, Birmingham, USA) using quartz cuvettes with 1 cm path lengths (105.253-QS, Hellma, Mühlheim, Germany). All spectra were corrected for background intensities by subtracting the spectra of pure solvent measured under identical conditions. Both excitation and emission bandwidths were set at 8.0 nm. The excitation wavelength at 295 nm was chosen, since it does not excite tyrosine residues. The emission spectrum was collected in the range of 300–500 nm with an increment of 1 nm. Each point on the emission spectrum is the average of 10 accumulations. The tested virus protease solutions (ZIKV NS2B/NS3^pro^, YFV NS2B/NS3^pro^, and CHIKV nsP2^pro^) had concentrations of 10 μM and the measuring volume was 50 µL. The buffer for the experiment contained 25 mM Tris-HCL, pH 8.5, 150 mM NaCl and 5% glycerol.

During the investigation of the ligand interaction with the targets, the protein solution within the cuvette was titrated stepwise with a ligand stock solution (0.5 mM ligand + 10 μM protein), ZIKV NS2B/NS3^pro^-riboflavin (0–70 µM), YFV NS2B/NS3^pro^-riboflavin (0–100 µM) and CHIKV nsP2^pro^-riboflavin (0–36 µM) and after each titration, a measurement was conducted. The quenching of the protease fluorescence, ΔF (F_max_ − F), at 330 nm of each titration point was used for fitting a saturation–binding curve using a nonlinear least-squares fit procedure, which has been discussed in detail elsewhere [49], based on Equation (2) [50].
Y = B_max_[Q]/K_D_ + [Q](2)
where [Q] is the ligand concentration in solution, acting as a quencher, Y is the specific binding derived by measuring fluorescence intensity, B_max_ is the maximum amount of the complex protease–ligand at saturation of the ligand and K_D_ is the equilibrium dissociation constant. The percentage of bound protease, i.e., Y, derived from the fluorescence intensity maximum, is plotted against the ligand concentration.

Additionally, the data were fitted with a modified Hill equation, obtaining the following relation (3) [51,52]:Log (F − F_min_)/(F) = m log K_D_ + n log [Q](3)
where F_min_ is the minimal fluorescence intensity in the presence of the ligand, K_D_ is the equilibrium constant for the protein–ligand complex. The “binding constant” K is defined as the reciprocal of K_D_.

The measurement of the thermal unfolding of the proteases with and without the ligand was performed in the range of 20–85 °C, with increments of 5 °C. The protein concentration was 10 µM and the riboflavin concentration 75 µM. The native protein fraction (fN) was calculated according to the following Equation (4):fN = (F − FU)/(FN − FU)(4)
where F, FN and FU are the steady-state fluorescence intensities of Trp at each temperature investigated, at the first (native protein) and the last (unfolded protein) temperature, respectively. The unfolded protein fraction can be calculated as follows: fU = 1 − fN. The temperature at which fN = fU is called melting temperature or transition midpoint (T_m_).

### 2.5. Molecular Dynamics and Computational Analysis

#### 2.5.1. Starting Structures

The atomic coordinates for CHIKV nsP2^pro^, YFV and ZIKV NS2B/NS3^pro^ were retrieved from the protein data bank (PDB) with the following codes: 3TRK [53], 6URV [54] and 5LCO [55], respectively. Then, 200 ns molecular dynamics (MD) simulations were carried out and clustering analyses were performed to obtain a stable and representative model for each of the three viral proteases. The stable protein model was used as a starting receptor for molecular docking calculations and subsequent MD simulations with riboflavin. Docking calculations were performed for the viral proteases with riboflavin using AutoDock Vina 1.1.12 [56]. The AutoDockTools program was used to add polar hydrogens and partial charges to the protein and rotational bonds in the ligands. The search space was defined in the gridbox near the binding site. As the output results, several poses were ranked according to the scoring function of Autodock Vina.

To prepare the ligand for MD simulations, Gaussian 09 [57] was used at the level of theory HF/6-31G* for geometry optimization and calculation of the electrostatic potential (ESP). After that, the antechamber [58] program was used to calculate the restrained electrostatic potential (RESP) charges and parmchk program was used to obtain the missing general amber force field (GAFF) [59] parameters.

#### 2.5.2. General Setup of the Molecular Dynamics Simulations

MD simulations were carried out using the Amber18 [60] software package. The all-atom protein interaction was described by the FF19SB [61] force field, while the riboflavin ligand interactions were described using GAFF and RESP charges. The protonation states of the amino acids side chain were set at pH 7.5 with H^++^ web server [62]. The systems were placed in an octahedral box, with TIP3P water extended 10 Å away from solute atoms and neutralized with Cl^−^ or Na^+^ ions. Each system was run thrice with the same starting structure and different atomic velocities. Initially, bad contacts were removed from the starting structure by two rounds of energy minimization. The first round was performed for 2500 steps of steepest descent, followed by 2500 conjugate gradient steps where the protein–ligand was constrained by the force constant of 10 kcal/mol-Å^2^, following which an unconstrained energy minimization was performed for 5000 steps. The systems were heated from 0.1 to 298 K under a constant atom number, volume and temperature (NVT ensemble), and the complex was restrained with a constant force of 10 kcal/mol-Å^2^. The equilibration process was conducted in six different steps, with constrained force constant decreasing from 10 to 0 kcal/mol-Å^2^, with 2 kcal/mol-Å^2^ per step over the protein–ligand atoms. Each step was carried out under a constant atom number, pressure and temperature (NpT ensemble) for 0.5 ns and 1 fs time step. Production run was performed for 200 ns with 2 fs time step in triplicate.

#### 2.5.3. Molecular Dynamics Analyses

The MD simulations were analyzed using the CPPTRAJ program [63] of the AmberTools19 package. The root mean square deviations (RMSD) were used to determine the equilibration of the systems and convergence of the simulations. For CHIKV and ZIKV, the entire protein was selected, while for the YFV, a part of the protein was selected, excluding 10 amino acids from the C-terminal of Chain A and from N-terminal of Chain B. Protein flexibility was examined by analyzing the root mean square fluctuation (RMSF) for the Cα atoms. The RMSF were calculated using the residue-by-residue over the equilibrated trajectories. Radius of gyration (RoG) and surface area were calculated to assess major structural changes in the protein. Average secondary structures over the simulation time for the proteins were assessed using the DSSP algorithm. Clustering analysis using k-means method, ranging from 2 to 6, was applied to each system to obtain the protein–ligand representative structure. The interaction energy was calculated using the generalized Born (GB)-Neck2 [64] implicit solvent model (igb = 8). Molecular mechanics/generalized Born surface area (MM/GBSA) energy was calculated between the protein and the ligand in the stable regime comprising the last 50 ns of the MD simulation, stripping all the solvent and ions.

### 2.6. Cell Line Cultures, Virus, and Reagents

Vero cells (Cercopithecus aethiops kidney normal; ATCC CCL-81) were grown in minimal essential medium (MEM) supplemented with 10% heat-inactivated fetal bovine serum (FBS), 100 U/mL^−1^ of penicillin, 0.1 mg/mL^−1^ of streptomycin and 0.5 µg/mL^−1^ of fungizone (Gibco, Waltham, MA, USA) and cultured at 37°C under a 5% CO_2_ atmosphere. C6/36 cells were maintained in Leibovitz-15 medium (L-15) with 10% FBS at 28 °C. Zika virus (strain ZIKVBR, [65]), fellow fever virus (vaccinal strain, Biomanguinhos) and Chikungunya virus (strain ECSA, [66]) stocks were cultured in C6/36 cells and titrated on Vero cells using plaque-forming assay.

### 2.7. Cytotoxicity Analysis

Briefly, 5 × 10^4^ Vero cells grown in 96-well plates in MEM were treated with riboflavin at a concentration range from 1000 to 0.48 µM for 48 h. Then, 1 mg/mL of 3-(4,5-dimethyl-2-thiazolyl)-2,5-diphenyltetrazolium bromide (MTT, Sigma, Aldrich, Saint Louis, MI, USA) was added to the cells, and they were incubated for 1h. Formazan crystals were dissolved in DMSO, and absorbance was determined at 550 nm using a microplate reader spectrophotometer (LT-4000 Microplatereader, LabTech). The results are shown as the percentage of viable cells relative to the untreated control cells. All assays were performed three times independently in quadruplicate. From this, the CC_50_ (cytotoxic concentration of the compound that reduced cell viability to 50%) was calculated from a dose–response curve in GraphPad Prism (version 8.00), using four-parameter curve-fitting.

### 2.8. Viral Infection Assays

To explore which step(s) of the viral lifecycle is blocked by the compound, time-of-drug addition experiments were performed. In brief, the substance was added to the virus and/or host cells at different time points relative to viral inoculation to the cells as follows: (1) pre-treatment of the virus with compound followed by inoculation of the treated virus to the cells and the virucidal or neutralizing activity of the caffeic acid is examined. (2) Viruses and compound were simultaneously added to cells during virus inoculation and the compound on the virus entry steps, including virucidal (neutralizing) activity and blockade of viral attachment and penetration to the cells, is examined. Viral infection experiments were performed in Vero cells seeded in 24-well plates treated with or without compound. In the different conditions reported above, the cells were infected with 50 PFU of virus for 1 h at 37 °C and revealed through the virus plaque-forming assay. Three independent experiments with quadruplicate measurements were performed.

### 2.9. Virus Plaque-Forming Assay

Briefly, Vero cells grown in 24-well culture plate were infected by 0.2 mL of ten-fold dilutions of supernatants. Following an incubation of 1 h at 37 °C, 0.5 mL of culture medium supplemented with 2% fetal bovine serum (FBS) and 1.5% carboxymethyl-cellulose sodium salt (Sigma-Aldrich, Saint-Quentin-Fallavier, France) were added, and the incubation was extended for 3 days at 37 °C. The cells were fixed (formaldehyde 10%) and stained with 2% crystal violet diluted in 20% ethanol, after the media had been removed. Plaques were counted and expressed as plaque-forming unit per milliliter (PFU·mL^−1^). The viral foci were counted to determine virus titer [67].

### 2.10. Statistical Analysis

All the experiments consist of at least two independent repetitions and all data are expressed as the means ± standard deviations (SDs). The statistical significances of the differences in the mean values were assessed with one-way analyses of variance (ANOVA), followed by Tukey’s multiple comparison test. For the viral infection assays and the plaque-forming assays, the statistical significances of the differences in the mean values were assessed with *t*-tests (Student’s t-distribution). Significant differences were considered at *p* < 0.05 (*), *p* < 0.01 (**) and *p* < 0.001 (***). All statistical analyses were performed with GraphPad Prism software version 8 (San Diego, CA, USA).

## 3. Results and Discussion

### 3.1. Expression and Purification of Flavivirus and Alphavirus Proteases

ZIKV NS2B/NS3^pro^, YFV NS2B/NS3^pro^ and CHIKV nsP2^pro^ were expressed in *E. coli* strains (BL21 (DE3) T1 (ZIKV and CHIKV) and Lemo (DE3) (YFV)) and purified with Ni-NTA sepharose and subsequent size exclusion chromatography to remove *E. coli* contaminates and aggregated protein species [36,37,44]. The SDS gels indicate the protease purity (Appendix A).

### 3.2. Inhibitory Potential of Riboflavin against Flavivirus and Alphavirus Proteases

Combating arboviruses infections is crucial due to the associated social and health implications. The identification of potential inhibitors that serve as platforms and lead compounds for the development efficacious and efficient alternatives to combat viral infection is one of the major priorities in countries at risk. The inhibitors need to possess compatible cellular activity, adequate pharmacokinetics and excellent oral bioavailability. Vitamins and their analogues possess these characteristics [17,68,69,70] and, in this context, the inhibitory effect of riboflavin was tested against ZIKV NS2B/NS3, YFV NS2B/NS3 and CHIKV nsP2 proteases in a primary inhibition test (Figure 1).

The primary test using 20 µM of riboflavin as the final concentration showed low inhibition of the ZIKV and YFV NS2B/NS3 proteases (10% and 30%, respectively); however, at the same concentration, the CHIKV nsP2^pro^ activity was inhibited 100% by riboflavin. The same experiment was performed for the CHIKV protease using 1 µM riboflavin, which showed inhibition of about 85%. This difference in the concentration needed to inhibit the proteases is probably related to the differences of the protease family and mode of interaction with the same molecule.

### 3.3. Characterization of Flavivirus and Alphavirus Proteases Inhibition by Riboflavin

The inhibitory effect of riboflavin on the ZIKV, YFV and CHIKV protease activities was further investigated at the concentration range of 0–240 µM (ZIKV), 0–200 µM (YFV) and 0–10 µM (CHIKV) (Appendix A). The final concentration to inhibit 100% of ZIKV NS2B/NS3^pro^ was settled at 220 µM (Appendix A) and YFV NS2B/NS3^pro^ at 200 µM (Appendix A). In contrast, the required amount of riboflavin to inhibit CHIKV nsP2^pro^ was a concentration of 5 µM (Appendix A), which showed the strong effect of the molecule against the nsP2 protease. Both enzymes from ZIKV and YFV are serine proteases; in contrast, CHIKV nsP2 is a cysteine protease, which clearly demonstrate the different IC_50_ values for riboflavin. By determination of the IC_50_ values, we observe the strongest effect of riboflavin inhibiting the nsP2 of CHIKV (93 ± 5.7 nM). On the other hand, the tested flavivirus proteases showed an IC_50_ around µM values of 79.4 ± 5.0 µM for ZIKV and 45.7 ± 2.9 μM for YFV (Figure 2 and Table 3).

Multiple biological functions of riboflavin have been described but to date, the inhibition of ZIKV, YFV, CHIKV or their isolated proteases have not been reported. However, the inhibition by riboflavin of the serine protease of the human cytomegalovirus (HCMV) was described with the determined IC_50_ value of 0.3 µM and a competitive inhibition mode of the protease [43]. Riboflavin has also been identified by computational studies as a potential inhibitor against the SARS-CoV-2 3CL protease (cysteine protease) [71]. These studies, together with our results, demonstrate the potential of riboflavin to inhibit viral proteases.

Classical Michaelis–Menten enzyme kinetic experiments determined the mechanism of inhibition of the tested proteases by riboflavin; the results revealed in Figure 2 that riboflavin is a competitive inhibitor of all three proteases. Characteristically of competitive inhibition is an unaffected maximum rate (V_max_) values for the enzymes in the presence of riboflavin in various concentrations; on the other hand, the Michaelis constant (K_m_) values increased (Figure 2B,D,F).

Ligand efficiency (LE) values validate the ratio of the ligand binding affinity versus the number of non-hydrogen atoms (N) of the molecule and for efficient binding, the values are typically > 0.30, with potential to be used as a lead structure for further development [47]. In the tested proteases, riboflavin possess a LE value of > 0.30 (0.36) against CHIKV nsP2^pro^, which confirmed the efficiency to bind to the active site of the CHIKV protease; conversely, the LE was below 0.30 for the flavivirus proteases (Table 3).

### 3.4. Investigation of the ZIKV, YFV NS2B/NS3^pro^ and CHIKV nsP2^pro^ Interaction with Riboflavin Using Tryptophan Fluorescence Spectroscopy

The riboflavin interaction with the flavivirus and alphavirus proteases was investigated using intrinsic tryptophan fluorescence spectroscopy (TFS). Based on the quenching of the proteases fluorescence of each titration point, a dissociation constant (K_D_) was determined for the proteases–riboflavin interaction (Figure 3).

The calculated K_D_ values are summarized in Table 3. Riboflavin shows a binding affinity of 2.8 ± 0.7 µM for CHIKV (Figure 3); it is about ten times stronger compared to the studied flavivirus proteases NS2B/NS3^pro^ (ZIKV—25.11 ± 2.0 μM) and (YFV—18.20 ± 3.9 μM) (Figure 3 and Table 4).

Riboflavin induces weak conformational changes in the examined protease structures, as observed by a blue edge excitation shift (BEES) of around 5 nm (335 to 330 nm) (Figure 3). Through BEES, the environment hydrophobicity of the tryptophan residue increases significantly [72].

The thermal denaturation of the single proteases (NS2B/NS3 of ZIKV and YFV) or (CHIKV nsP2), as well as the enzymes in complexes with riboflavin, was followed by fluorescence spectroscopy (Appendix A). The ZIKV NS2B/NS3^pro^–riboflavin complex showed about +8 °C change in the melting temperature (ΔTm) (Appendix A and Table 4), while CHIKV and YFV proteases shows a Tm of +5 °C and +4 °C, respectively, as shown in Appendix A and Table 4. The greater thermal stability of the ZIKV NS2B/NS3 protease riboflavin complex compared to the other proteases may be related to the general folding energetic requirements, which increases the numbers of hydrogen bonds and results in better internal van der Waals’ packing in the tertiary structure.

### 3.5. Interaction of Riboflavin with Flavivirus NS2B/NS3^pro^ and Alphavirus nsP2^pro^ Explored by MD Simulations

Computational techniques have been applied in the drug discovery and development pipeline to allow simulating the dynamic nature of the binding event. Using molecular docking, we anchor the riboflavin molecule directly in the active site of ZIKV NS2B/NS3^pro^ (PDB id: 5LC0), YFV NS2B/NS3^pro^ (PDB id: 6URV) and CHIKV nsP2^pro^ (PDB id: 3TRK) structures, as the molecules shows a competitive inhibition mode, which is determined experimentally (Figure 2). Subsequently, MD simulations were performed and indicated the presence of stable interactions formed between the ligand and the proteins. Three sets of simulations (200 ns each) were performed independently, based on binding energy and frames in the cluster; a representative structure was chosen for each protease–ligand complex (Table 5, Appendix A).

Furthermore, these riboflavin–proteases complexes were subject to RMSD, RMSF, RoG and the surface area analysis using 200 ns performed in triplicate for each complex (Appendix A). The position of riboflavin in the active site of the three proteases is presented in Figure 4.

The decomposition analysis of the total binding energy provides the contribution to the binding of each amino acid (Figure 4). What is outstanding in the interaction of the virus proteases with riboflavin is the region surrounding the active site residues of the proteases. The amino acids with ΔG_binding_ values lower than 1.0 kcal mol^−1^ are His51, Asp75, Ser135 (ZIKV), His53, Asp77, Ser138 (YFV), Cys11 and His81 (CHIKV). Remarkably, riboflavin interacts with two amino acids of the catalytic triad of ZIKV NS2B/NS3^pro^ (His51 and Ser135) (Figure 4A). All amino acids are numbered according to the PDB model.

In the case of YFV NS2B/NS3^pro^, Asp77 of the active site triad and the neighboring residues Lys75 and Glu76 are involved in the binding (Figure 4B). Residues of the NS2B cofactor of the flavivirus proteases are also involved in the interaction with riboflavin (ZIKV Asp 37 and YFV Leu30, Glu35) (Figure 4A,B).

The catalytically important Cys11 residue of CHIKV nsP2^pro^ interacts directly with riboflavin, as well as the region surrounding His81 (Tyr77, Asn80 and Trp82). The three independent MD simulation replicas indicated a very similar location for riboflavin in the three proteases structures (Appendix A). The interacting amino acid residues were all the same with minor intensity changes in the energy, due to statistical fluctuations over the trajectory. Amino acid residues that stabilize the interaction with riboflavin by hydrogen bonds and hydrophobic interaction are shown in Appendix A.

The analysis of the interactions occurring at the proteases’ subsites (S1, S1′, S2, S3 and S4) provides valuable information for future optimizations. Therefore, a detailed structural representation of the complexes was included in Figure 5. The amino acids comprising the substrate binding sites of ZIKV and YFV NS2B/NS3^pro^, and CHIKV nsP2^pro^ have been described previously [35,53,54,73] (Appendix A). Riboflavin extends along with the S3 and S4 subsites of the ZIKV protease. Structural representation of the interfaces of YFV protease shows that the riboflavin interacts mostly with the S1′ and S2 subsites. As can be observed in the Figure 5, riboflavin engages with the CHIKV protease through interaction with the S1 and S4 subsites.

Riboflavin not only sits in the substrate-binding region of the proteases, it also restricts the access of the substrate to the active site. In addition, when riboflavin fits optimally in this area, the binding energy of the cluster, IC_50_ and K_D_ present feasible values.

In the case of the YFV NS2B/NS3^pro^, riboflavin interact mainly with residues of the S1’ and S2 sites, which also may explain the IC_50_ value of 45.7 ± 2.9 μM. The effect of riboflavin against CHIKV nsP2^pro^ seems to be the greatest. The inhibitor blocks access to the S1 and S4 subsites and retains the highest binding energy of all systems (−40.7 ± 6.2 kcal/mol) (Table 4), which is in agreement with the results of the experiments described before, where the IC_50_ value is in the nM range and the K_D_ value in the lower µM value.

### 3.6. Cytotoxicity and Antiviral Effect of Riboflavin against Virus Infected Cell Cultures

To assess the effect of riboflavin on cell viability and virus infection, we performed e 3-[4,5-dimethylthiazol-2-yl]-2,5-diphenyltetrazolium bromide (MTT) assay at a concentration ranging from 1000 to 0.48 µM for 48h. The percentages calculated for the different sample concentrations were plotted and the mean cytotoxic concentration values (CC50) were estimated to be 72.31 µM. (Appendix A). Time-of-addition type of experiments were used to analyze the effect of riboflavin on different stages of the viruses’ replication. For all of these assays, cells were treated with 50 µM of the compound, a concentration that does not affect cell viability.

To explore the antiviral activity of riboflavin, we analyzed whether the compound inhibited virus plaque formation. The formation of plaques corresponds to a full infection cycle, from entry to egress of newly formed viral particles. To explore which step(s) of the viral lifecycle is blocked by the molecule, time-of-drug addition experiments were performed. The inhibitory effect of the compound on the viruses’ expressions was assessed in virucidal assay against CHIKV, which decreased 44.35% (±12.49) of CHIKV replication (*p* < 0.001), indicating that this compound strongly inhibited the CHIKV entry (Figure 6). As we did not use the viral replicon assay, which will target the genes of the non-structural proteins necessary for replication, we can only infer that riboflavin also interacts with structural proteins of the Chikungunya virus, preventing its entry, which was not observed for ZIKV and YFV. We can suggest that riboflavin has only one therapeutic target in ZIKV and YFV viruses. On the other hand, CHIKV can be affected by the vitamin through the entry phase, as well as in the replication phase.

Our results indicate that riboflavin significant reduced CHIKV virus entry to the host cells at non-toxic concentrations. The strong virucidal effect observed for riboflavin in CHIKV might suggest that an anti-CHIKV mechanism of action for this complex might be related to a direct action on the viral particle envelope [74,75,76], which could also be responsible for the effect observed on virus attachment [77,78]. The same effect does not appear for the other tested viruses, which could be a reasonable explanation for the observed virucidal effect and the virus’s family selectivity.

## 4. Conclusions

Neurotropic flavi- and alphaviruses can cause simultaneous infections in the cases of ZIKV, YFV and CHIKV co-infections by two or all these viruses, which has been previously reported [79,80,81,82,83,84,85]. Unfortunately, there are relatively few prophylactics or therapeutics treatments for these viruses and they are mostly very specific for one virus-associated target and generally developed for a single virus or viral strain. In the case of the possible flavi- and alphavirus co-infections, broad-spectrum antivirals are needed, which inhibit the replication of a wide range of viruses. Overall, broad-spectrum antiviral drugs are more difficult to design compared to classical virus-specific drugs, due to the differences among viruses, not only in their structures, but also in their modes of infecting the host. Our results demonstrate that riboflavin can inhibit flavivirus proteases in a moderate µM range and alphavirus protease in a moderate nM range. An improvement process using the riboflavin core structure as a lead molecule will contribute to enhanced binding and inhibition properties against the flavivirus proteases, in terms of not reducing the inhibitory potential against the alphavirus protease. In drug development, this improvement process is an essential procedure and can include chemical modification of the riboflavin core and investigation of the related flavin analogues.

## Figures and Tables

**Figure 1 microorganisms-10-01331-f001:**
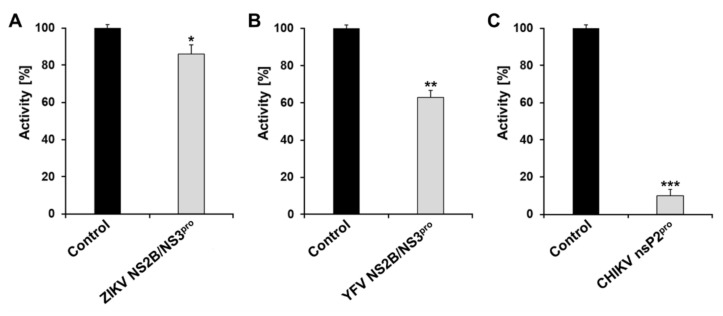
Primary inhibition tests of riboflavin against arboviruses protease through enzymatic activity [%]. A total of 20 µM of riboflavin was tested against (**A**) ZIKV NS2B/NS3^pro^ (3 nM), (**B**) YFV NS2B/NS3^pro^ (6 nM) and 1 µM of riboflavin against (**C**) CHIKV nsP2^pro^ (1 μM). Asterisks mean that the data differs from the control (0 µM inhibitor) significantly at *p* < 0.05 (*), *p* < 0.01 (**), and *p* < 0.001 (***) level according to ANOVA and Tukey’s test. Data shown are the mean ± SD from three independent measurements (*n* = 3).

**Figure 2 microorganisms-10-01331-f002:**
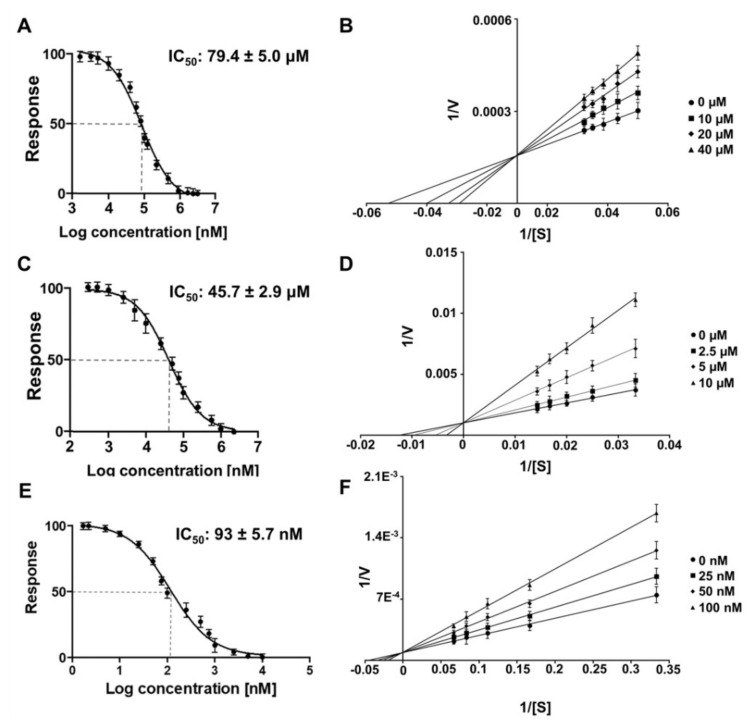
ZIKV and YFV NS2B/NS3^pro^ and CHIKV nsP2^pro^ activity test under riboflavin influence and inhibition mode. Dose response curves for IC_50_ determination. The normalized response (%) of the proteases is plotted against the log of the riboflavin concentration. The determined IC_50_ values are presented within the corresponding picture. Lineweaver–Burk plots for determining the inhibition mode. [S] is the substrate concentration; v is the initial reaction rate. Data shown are the mean ± SD from three independent measurements (*n* = 3). (**A**) Dose response curve of riboflavin and ZIKV NS2B/NS3^pro^. (**B**) Lineweaver–Burk plot of riboflavin and ZIKV NS2B/NS3^pro^. (**C**) Dose response curve of riboflavin and YFV NS2B/NS3^pro^. (**D**) Lineweaver–Burk plot of riboflavin and YFV NS2B/NS3^pro^. (**E**) Dose response curve of riboflavin and CHIKV nsP2^pro^. (**F**) Lineweaver–Burk plot of riboflavin and CHIKV nsP2^pro^.

**Figure 3 microorganisms-10-01331-f003:**
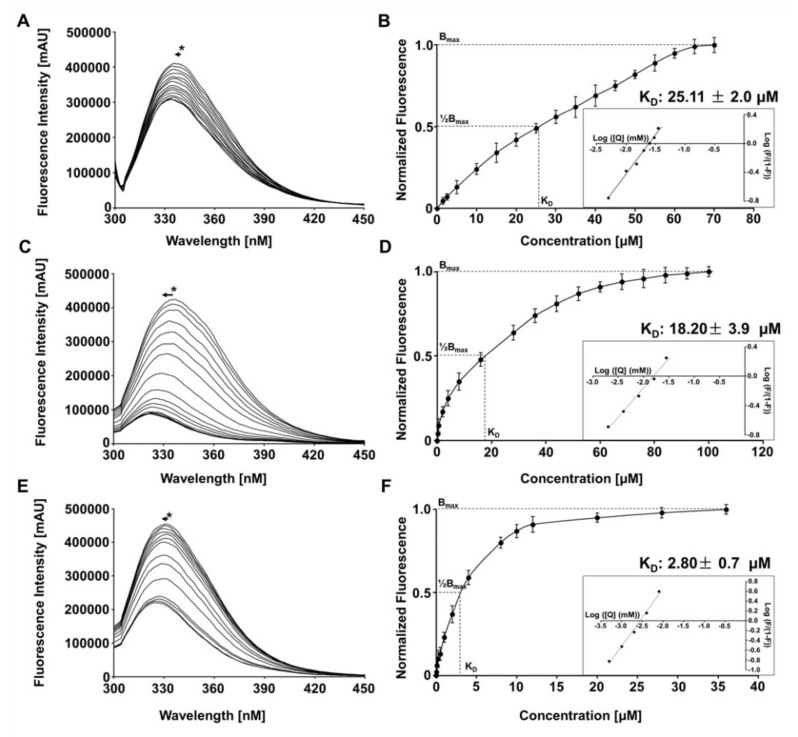
Fluorescence spectroscopy of Trp at 295 nm of ZIKV, YFV and CHIKV protease in the presence of riboflavin. (**A**) Fluorescence of ZIKV NS2B/NS3^pro^ under influence of riboflavin titration. (**B**) Binding saturation curve determined a K_D_ value of 25.1 ± 2.0 µM for the ZIKV NS2B/NS3^pro^–riboflavin interaction. K_D_ determination using a modified Hill equation, intersection with x-axis corresponds to the logarithmic value of the K_D_. (**C**) Fluorescence of YFV NS2B/NS3^pro^ under influence of riboflavin titration. (**D**) Binding saturation curve determined a K_D_ value of 18.2 ± 3.9 µM for the YFV NS2B/NS3^pro^–riboflavin interaction and K_D_ determination using a modified Hill equation. (**E**) Fluorescence of CHIKV nsP2^pro^ under influence of riboflavin titration. (**F**) Binding saturation curve determined a K_D_ value of 2.8 ± 0.7 µM for the CHIKV nsP2^pro^–riboflavin interaction and K_D_ determination using a modified Hill equation. Data shown are the mean ± SD from three independent measurements (*n* = 3). Asterisks demonstrate a blue excitation shift of visible Trp.

**Figure 4 microorganisms-10-01331-f004:**
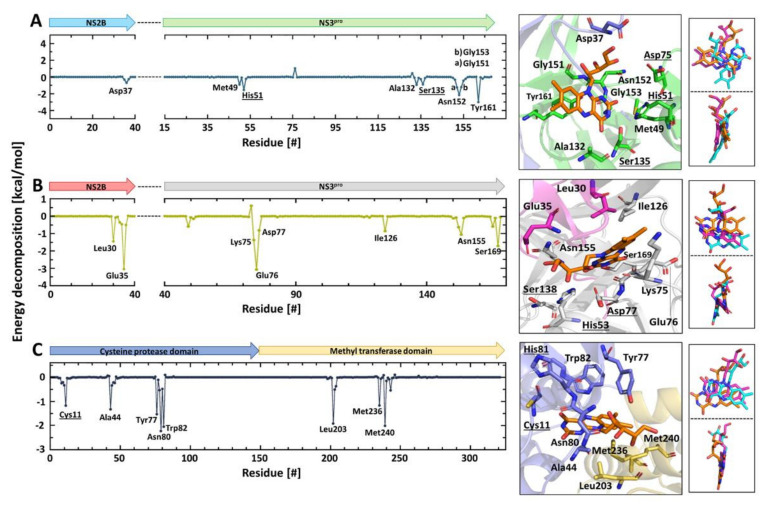
Amino acids contributing in the ZIKV NS2B/NS3^pro^–, YFV NS2B/NS3^pro^– and CHIKV nsP2^pro^–riboflavin interaction. Decomposition energy of the amino acids involved in the interaction with riboflavin based on the MD simulations. Coordination of riboflavin in the protease-binding region and overlay of the riboflavin replicate 1, 2 and 3. (**A**) ZIKV NS2B/NS3^pro^–riboflavin interaction. (**B**) YFV NS2B/NS3^pro^–riboflavin interaction. (**C**) CHIKV nsP2^pro^–riboflavin interaction. Active site residues are underlined.

**Figure 5 microorganisms-10-01331-f005:**
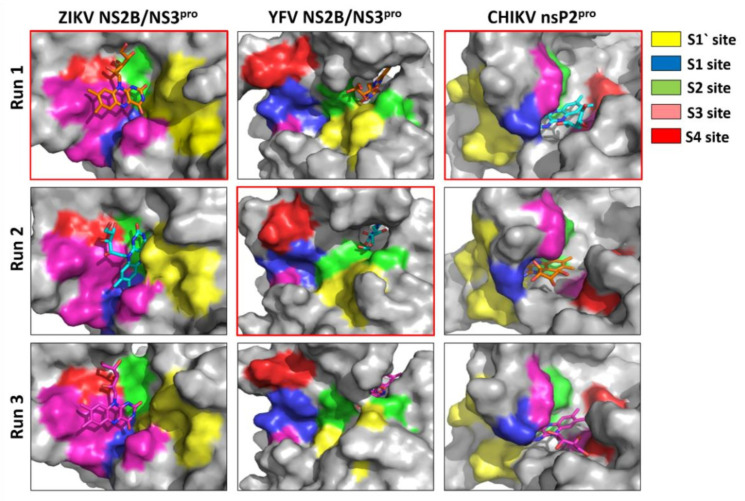
Substrate binding pockets (surface representation) of ZIKV NS2B/NS3^pro^, YFV NS2B/NS3^pro^ and CHIKV nsP2^pro^ (S1, S1′, S2, S3, S4) under the influence of riboflavin. The results of the representative structure of each simulation replica is shown to demonstrate the reproducibility of our results. The inhibitor is shown in sticks.

**Figure 6 microorganisms-10-01331-f006:**
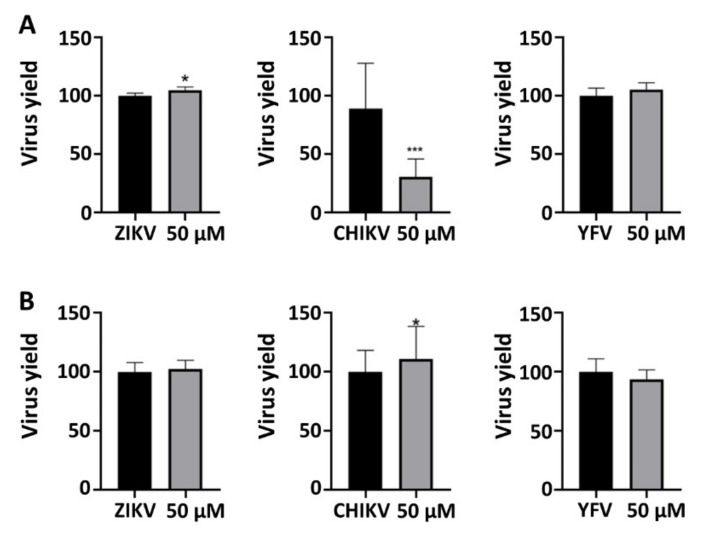
Virus plaque-forming assay. (**A**) Vero cells were infected with 50 PFU of viruses and treated with the compound at 50 μM for 72 h. (**B**) Viruses and riboflavin were incubated for 1h at RT and then for one additional hour in the cells. Then, the compound was removed, and the cells were added to media. Viruses plaque-forming were counted to compare the virus replication percent. * *p* < 0.05 and *** *p* < 0.001 was considered significant.

**Table 1 microorganisms-10-01331-t001:** Examples of neurotropic viruses in the flavivirus and alphavirus families.

Family	Virus	Neurological Manifestation	Reference
Flaviviridae	West Nile virus (WNV)	Meningoencephalitis	[7]
	Yellow fever virus (YFV)	Fatal encephalitis	[8]
	Japanese encephalitis virus (JEV)	Japanese encephalitis	[9]
	Saint Louis encephalitis virus (SLEV)	St. Louis encephalitis	[10]
	Tick-borne encephalitis virus (TBEV)	Tick-borne encephalitis	[11]
	Zika virus (ZIKV)	Microcephaly, Guillain–Barré syndrome	[12,13]
Alphaviridae	Chikungunya virus (CHIKV)	Encephalopathy, neuropathy, myelopathy, encephalomyelitis	[14]
	Sindbis virus (SINV)	Encephalomyelitis	[15]
	Venezuelan equine encephalitis virus (VEEV)	Encephalomyelitis	[16]
	Semliki forest virus	Encephalitis	[16]

**Table 2 microorganisms-10-01331-t002:** Examples of NS2B/NS3 and nsP2 protease inhibitors.

Compound	Targeted Virus	IC_50_ (µM)	Reference
Novobiocin	ZIKV	14.2	[30]
Simeprevir	ZIKV	2.6	[31]
Methylene blue	ZIKV	-	[32]
Niclosamide derivative	ZIKV	-	[33]
Erythrosin B	ZIKV	-	[34]
	YFV	-	[34]
Peptides	YFV	0.05–>125	[35]
Hesperetin	YFV	2.0	[36]
	ZIKV	12.6	[37]
	CHIKV	2.5	[37]
1,3-Thiazolbenzamide derivatives	CHIKV	13.1 and 8.3	[38]
QVIR	CHIKV	-	[39]

**Table 3 microorganisms-10-01331-t003:** Inhibitor key numbers and inhibition mode for riboflavin ZIKV, YFV NS2B/NS3^pro^ and CHIKV nsP2^pro^.

Protease	IC_50_ ± STD	Inhibition Mode	*p*IC_50_^1^	LE ^2^	N ^3^	LogP ^4^
CHIKV nsP2	93 ± 5.7 nM	competitive	7.0	0.36	27	−1.5
YFV NS2B/NS3	45.7 ± 2.9 μM	competitive	4.3	0.22	27	−1.5
ZIKV NS2B/NS3	79.4 ± 5.0 μM	competitive	4.1	0.21	27	−1.5

^1^ Logarithm of IC_50_ value (*p*IC_50_). ^2^ LE: ligand efficiency; LE > 0.3 suggests that the molecule is a potent lead compound. ^3^ Number of non-hydrogen atoms (N). ^4^ Information about hydrophobicity of the molecule (https://www.drugbank.ca/drugs/DB00140, accessed on 20 March 2022). A negative value for logP means that the compound has a higher affinity for the aqueous phase (it is more hydrophilic). When LogP = 0, the compound is equally partitioned between the lipid and aqueous phases; a positive value for LogP denotes a higher concentration in the lipid phase (i.e., the compound is more lipophilic).

**Table 4 microorganisms-10-01331-t004:** Summary of the fluorescence spectroscopy experiment results of ZIKV and YFV NS2B/NS3^pro^ and CHIKV nsP2^pro^ with riboflavin.

Protease	K_D_ ± STD	ΔTm	Excitation Shift
CHIKV nsP2	2.80 ± 0.7 μM	5 °C	blue edge excitation shift
YFV NS2B/NS3	18.20 ± 3.9 μM	4 °C	blue edge excitation shift
ZIKV NS2B/NS3	25.11 ± 2.0 μM	8 °C	blue edge excitation shift

**Table 5 microorganisms-10-01331-t005:** Summary of the MD simulation replicas 1, 2 and 3 of ZIKV, YFV and CHIKV proteases in complex with riboflavin. Chosen representative protease–ligand structure in bold.

Protease + Riboflavin	Replicate	Binding Energy (kcal/mol)	Representative Frames Cluster (%)
ZIKV NS2B/NS3	**1**	**−34.8 ± 4.5**	**75.5**
	2	−27.9 ± 4.2	81.1
	3	−28.2 ± 6.6	87.5
YFV NS2B/NS3	1	−28.7 ± 5.7	91.6
	**2**	**−40.6 ± 5.0**	**79.5**
	3	−32.5 ± 5.1	92.7
CHIKV nsP2	**1**	**−40.7 ± 6.2**	**56.7**
	2	−28.5 ± 6.0	78.1
	3	−34.6 ± 11.5	71.3

## Data Availability

All data are reported in the text and Appendix A; additional data that support the findings of this study are available upon request from the corresponding authors, M.A.C. and R.J.E.

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
