# Peer review of "Riboflavin, a Potent Neuroprotective Vitamin: Focus on Flavivirus and Alphavirus Proteases"

_microorganisms, 2022, doi:10.3390/microorganisms10071331_

Round 1

Reviewer 1 Report

The Manuscript titled “Riboflavin, a Potent Neuroprotective Vitamin: Focus on flavivirus and alphavirus proteases” which discuss the inhibitory potential of the water-soluble vitamin riboflavin against NS2B/NS3pro of ZIKV and YFV and, nsP2pro of CHIKV is of great importance. Overall manuscript is written and presentation well.

Major comment:

Cell based antiviral data must be included to prove the antiviral potential of Riboflavin against ZIKA, YFV and CHIKV.

Minor comment:

1.    Authors should be consistent in writing. For e.g., in the below sentence, somewhere word or number is used.

Three nM ZIKV NS2B/NS3pro was pre-incubated with riboflavin at different concentrations for 60 min at RT (0-40 μM). CHIKV nsP2pro at one μM was incubated with 0-100 nM riboflavin and YFV NS2B/NS3pro at 6 nM with 0-10 μM riboflavin

2.    The atomic coordinates for CHIKV nsP2pro, YFV and ZIKV NS2B/NS3pro was retrieved from PDB data bank with the respective codes: 3TRK, 6URV and 5LCO. - Plz add references for these PDB file.

3.    ZIKV NS2BNS3pro, YFV NS2BNS3pro and CHIKV nsP2pro were expressed in E. coli Strains. Which strains of E. Coli was used? BL21, Rosetta or any other? 

4.    Combating arboviruses infections are of crucial importance due to the associated so- Remove word importance in this sentence. 

5.    Make sure in figure 2 A and C, X-axis is Log Concentration nM. It looks like it should be µM. In figure 2A and C, Y-axis can start from log concentration 3nM 

6.    Introductions need more detail. Authors should mention about recent development in protease inhibitor against these viruses.

Authors can site following paper in your manuscript.

In vitro and in vivo characterization of erythrosin B and derivatives against Zika virus 

Antiviral Agents against Flavivirus Protease: Prospect and Future Direction

 Methylene blue is a potent and broad-spectrum inhibitor against Zika virus in vitro and in vivo

 JMX0207, a Niclosamide Derivative with Improved Pharmacokinetics, Suppresses Zika Virus Infection Both In Vitro and In Vivo

Reviewer 2 Report

Reference manuscript Microorganisms 2022: Riboflavin, a Potent Neuroprotective Vitamin: Focus on flavivirus and alphavirus proteases.  Raphael J. Eberle et al., 2022.

General Comments:

In the presented text, the authors are reporting their results of studies of the inhibitory potential of Riboflavin on strains of chikungunya virus (Alphavirus genus), zika virus and the yellow fever virus ( Flavivirus). Throughout the text, the authors show by using inhibitory experiments, Molecular Docking and Biophysical evaluations that riboflavin has a competitive inhibitory action on viral proteases, suggesting the potential use of this vitamin as an antiviral drug or blocker of viral infections. The text is clear, scientifically coherent, showing that the authors have expertise in the subject. Viral infections are currently emerging issues, indicating enormous potential for different viruses to bring irreversible problems to humanity. After careful reading, it is my opinion that the text has scientific merit, but in a revised version, when the authors will be able to add suggestions that will bring greater interest to readers in the area.

Comments

1-  Please, when authors use abbreviations, as in the abstract of this text, lines 22 and 23, and other parts of text, the first time the citation appears in the text must be followed by the definition of these abbreviations!  This is a standard procedure in scientific texts.   2- Between lines 77 and 78, the authors wrote … competitively bound to the active site of the tested proteases with… Here it would be interesting for the authors to indicate the proteases studied and the reason for their choices. It would also be interesting the authors did not save the text and indicate the classes of proteases evaluated (metallo, serine, cysteine, aspartate).   3- In line 237, when the authors detail the obtainment of recombinant viral proteases and indicate the data as supplementary material, in my opinion I would put this data as figure 1 of the manuscript. Although the expressions are not the main objective of the group, these molecules are the targets of studies of the inhibitory agent and this already configures them as important. In the figure the authors can show only the purified recombinant forms.   4- I would also make some corrections in the figure legend about the purifications of recombinant proteases. Replace SDS Gel with SDS-PAGE, Place Molecular Mass Markers instead of protein marker, place kDa on left top of the gels. Detail the procedures in materials and methods, explain whether the electrophoresis was performed under reducing conditions or not? And finally explain why the authors show two lanes for the purified proteins? Affinity chromatography and molecular exclusion.   5- Text written between lines 244 to 249 could be placed in the manuscript introduction. Undoubtedly it is important to relate to the experimental data, but it is not characterized as a result.   6- In the figure 1 of the manuscript, lines 251 to 255, where the authors report protease inhibition experiments. Authors could better detail the data in the legend! What concentrations of proteases were used? Why they used only one concentration of inhibitor per enzyme, instead of doing inhibition kinetics?   7- In my opinion, figure S3 in the supplementary material is more comprehensive and elegant than figure 1 in the manuscript. I suggest the authors make the exchanges!   8- The text written between lines 289 to 295 is not characterized as being results and could be added in the introduction of the manuscript.   9- Once again the figure shown in the supplementary material as S4 could be in the main body of the manuscript, along with Table 3. I don't know why the authors are producing a text in the results chapter in a summarized form, when they could enrich their experimental data!   10- On line 356, the letter S is in red. Make correction!   11- The Molecular Docking data indicating the potential of riboflavin to interact with the different viral proteases studied, and shown in figure 3 of the manuscript and S6 to S11 in the supplementary material are very interesting, especially because they were made based on crystallography of recombinant viral proteases. This increases confidence in these results!   12- But did the authors do experiments based on Molecular Dynamics and Quantum Chemistry showing the distances between the amino acid residues pointed out as important in these molecular interactions, which proves that these amino acids are really interacting with riboflavin?   13- Also in this sense of a final proof of these molecular interactions among viral proteases and riboflavin, with amino acids involved. Did the authors produce mutated isoforms of these amino acids, which could inhibit the inhibitory activity of riboflavin? That would be elegant and very important to final proof!   14- The main criticism in the presented manuscript is that the authors do not show a correlation of their data generated through biochemical and biophysical assessments with biological assessments. At no time did the authors make, but should have made, an attempt by using riboflavin to inhibit viral growth cycles in cells maintained in culture, for example, or even in models using infected animals. Without these data, we have pioneering data, which simply lack biological proof.   15- Still in this example of viruses with neuronal tropism, I would like to know if the authors at some point thought of studying the effects of Riboflavin on the rabies virus, a condition that leads to the death of practically all infected patients, if not treated in time with specific serum therapy.

Round 2

Reviewer 1 Report

Authors has corrected most f the suggestion given by reviewer. 

Some minor point

1. Which vero cells has been used? Vero E6 cells? 

2.  5 × 104 Vero cells grown in 96 line (264 section 2.7. ) superscript is

missing. 

3. None of the suggested references has been cited!!!
